# Contamination of medical devices and hospital environments with free-living amoebae: Evidence from hospitals in Northwestern Iran

Hamed Behniafar[1], Adel Spotin[2], Ali Bahadori[3], Soghra Valizadeh[4], Leili Valizadeh[5], Hosein Pazoki[6], Fatemeh Mahdavi[7]☯*, Maryam Niyyati[8]☯*

**1** Department of Medical Parasitology, Sarab Faculty of Medical Sciences, Sarab, East Azerbaijan, Iran, **2** Department of Parasitology and Mycology, Faculty of Medicine, Tabriz University of Medical Sciences, Tabriz, Iran, **3** Department of Medical Microbiology, Sarab Faculty of Medical Sciences, Sarab, East Azerbaijan, Iran, **4** Department of Food Hygiene and Aquatic, Faculty of Veterinary Medicine, University of Tabriz, Tabriz, Iran, **5** Department of Cardiology, School of Medicine, Ardabil University of Medical Sciences, Ardabil, Iran, **6** Department of Medical Microbiology, Faculty of Medicine, Infectious Diseases Research Center, Gonabad University of Medical Science, Gonabad, Iran, **7** Department of Medical Parasitology and Mycology, Urmia University of Medical Sciences, Urmia, Iran, **8** Department of Medical Parasitology and Mycology, Faculty of Medicine, Shahid Beheshti University of Medical Sciences, Tehran, Iran

☯ These authors contributed equally to this work.
* fmahdaviabhari@gmail.com (FM), maryamniyati@yahoo.com (MN)

## Abstract

Free-living amoebae (FLAs) are ubiquitous protozoa found in soil, air, and artificial systems, including hospital environments. Some genera of free-living amoebae, such as *Acanthamoeba*, can cause severe health complications, including *Acanthamoeba* keratitis and granulomatous amoebic encephalitis. This study investigated the presence of free-living amoebae (FLAs) in hospital environments, including ready-to-use medical devices and equipment (such as lasers, swabs, and forceps), as well as beds and gowns. To the best of our knowledge, FLAs in these medical devices and equipment have been examined for the first time. In this cross-sectional study, 45 environmental and medical device samples were collected from two hospitals in Northwestern Iran. After filtration, the samples were cultured in a 1.5% non-nutrient agar medium enriched with *Escherichia coli*. The growth of the FLAs and their genera was determined through microscopic analysis. Positive samples were submitted for PCR analysis targeting the 18S rRNA gene, followed by sequencing and phylogenetic analysis. Also, the pathogenicity of *Acanthamoeba* isolates was evaluated through osmo- and thermotolerance tests. FLAs were detected in 22.22% (10/45) of samples using microscopy. Most of the examined sources (90%) had mixed contamination, including *Acanthamoeba*, *Vahlkampfia*, and *Veramoeba* (4), *Acanthamoeba* and *Vahlkampfia* (1), *Acanthamoeba* and *Veramoeba* (2), and *Veramoeba* and *Vahlkampfia* (2). Also, one source showed sole contamination with *Vahlkampfia*. Among the positive samples, 5 were obtained from environmental sources, 4 from equipment, and 1 from surgical gowns. Most *Acanthamoeba* isolates demonstrated

**Data availability statement:** All relevant data are within the paper.

**Funding:** The funders (Sarab Faculty of Medical Sciences, Grant No.: 401000013) had no role in study design, data collection and analysis, decision to publish, or preparation of the manuscript.

**Competing interests:** The authors declare no competing interests.

osmo-tolerance (72.48% at 0.5 M) and thermo-tolerance (100% at 37°C).. Sequence analysis identified *Acanthamoeba* T4 genotype (5), *Vahlkampfia* sp. (3), and *V. vermiformis* (6). In this study, FLAs were isolated from patients' beds and surgical gowns for the first time, emphasizing new infection risks within an ophthalmology hospital. In addition to the high prevalence of FLAs in the examined sources, most *Acanthamoeba* isolates were found to be resistant to osmotic stress and heat shock, which supports their pathogenic potential. However, these findings highlight the need for improved disinfection protocols for sterile equipment.

## Introduction

Free-living amoebae (FLAs) are protozoa that are ubiquitous in various environments, including soil, air, and artificial systems such as hospital water networks and air conditioning units [1]. These amphizoic organisms exist as either trophozoites or resistant cysts, enabling them to survive under harsh conditions. *Acanthamoeba* cysts, as one of the most clinically important genera of FLAs, exhibit pronounced tolerance to many widely used disinfectants because their sturdy, double-layered cyst wall (comprising an outer protein–polysaccharide layer and an inner cellulose-enriched layer) substantially restricts penetration of biocidal agents [2]. While most FLAs are harmless and vital for ecosystem functions, such as organic decomposition and nutrient recycling [3,4], particular species or genotypes from genera like *Acanthamoeba*, *Naegleria*, *Balamuthia*, *Sappinia*, and *Vermamoeba* (formerly *Hartmannella*) pose health threats [5]. *Acanthamoeba* and *Naegleria* have garnered significant research interest due to their links to severe diseases, including *Acanthamoeba* keratitis (AK), granulomatous amoebic encephalitis (GAE), and primary amoebic meningoencephalitis (PAM) [6].

Instead of using traditional morphological names such as *Acanthamoeba polyphaga*, molecular characterization now enables the classification of *Acanthamoeba* isolates into genotypes (T1–T23). T4 predominates in both environmental and clinical samples [7]. AK, a vision-threatening corneal infection caused by *Acanthamoeba*, is associated with risk factors such as the use of soft contact lenses, corneal injury, and exposure to contaminated water. In Iran, AK incidence has risen recently, with T4 implicated in most cases [8]. GAE, a fatal CNS infection, mainly affects immunocompromised patients, such as those with HIV/AIDS or on chemotherapy [9]. PAM, caused by *Naegleria fowleri*, is rarer but nearly always lethal, often acquired via nasal exposure to contaminated warm water during activities like irrigation or swimming [10]. FLAs also act as reservoirs for bacteria, fungi, and viruses, thereby amplifying the risks of nosocomial infections [11].

Immunocompromised individuals, those with HIV/AIDS, cancer, or receiving corticosteroids, are particularly vulnerable to FLA infections [12]. High-risk settings include ICUs and ophthalmology wards, where patients face surgical procedures amid potential contamination from water systems and equipment [13]. Standard disinfection protocols often fail to eliminate *Acanthamoeba* cysts on devices such as

oxygen masks, dialysis units, and air conditioners [14,15]. Given the high admission rate of patients with corneal injuries at the selected Ophthalmology Hospital (Tabriz Alavi Hospital), and considering that many are contact lens users or post-operative cases, the risk of AK is elevated, necessitating detailed research.

Iranian studies have detected FLAs in hospital dust, therapeutic pools, and municipal water [16,17]. A 2018 analysis of ICU nasal swabs and dust samples revealed a 28% prevalence of FLAs, primarily caused by *Acanthamoeba* spp. [14]. Yet, data on FLAs in specialized facilities like ophthalmology centers are scarce. The chosen ophthalmology hospital in northwestern Iran, handling numerous suspected AK referrals, lacks prior environmental FLA assessments. Likewise, the selected general hospital, which serves immunocompromised patients, has no known prevalence data for FLAs.

This study examined the presence of FLAs in these two hospitals using both microscopic and molecular methods. Unlike prior research on hospital FLAs, it uniquely targets an ophthalmology setting and scrutinizes ready-to-use surgical tools, including sterile instruments.

## Materials and methods

### Ethical approval

The study received approval from the Ethics Committee of the Sarab Faculty of Medical Sciences, under the code IR.SARAB.REC.1402.02.

### Sample collection

In this cross-sectional study, a total of 45 samples were collected from two hospitals in East Azerbaijan Province, North-western Iran: 30 samples from the Ophthalmology Hospital on November 19, 2023, and 15 samples from the General Hospital on December 13, 2023 (S1 Table). Sample size based on the expected 20–30% prevalence from prior studies; unequal due to focus on a high-risk ophthalmology hospital. Sampling focused on two categories: environmental speci-mens and sterilizable medical equipment. All samples were transported at room temperature within 24 hours to the Parasi-tology Laboratory at Sarab Faculty of Medical Sciences for analysis.

**The ophthalmology hospital samples.** Due to the importance of eye trauma for developing AK, two-thirds of the samples (30 samples) were collected from the specialized eye hospital. These included 18 environmental specimens, such as dust collected from high-contact areas (e.g., hallways, patient beds, ventilation systems), and 12 equipment samples from sterilized medical instruments (e.g., lasers, swabs, forceps) that were prepared for ophthalmic procedures.

**The general hospital samples.** All 15 samples from this site were collected from medical devices, such as angiocatheters, scalpels, surgical gowns, and protective shields, which are commonly used in clinical settings.

### Isolation and morphological identification

Environmental swabs and equipment samples were immersed in 200 mL of sterile water for 24 hours. Suspensions were filtered through 1.6 µm pore-size cellulose nitrate membranes [18]. The central 2 cm portion of each membrane was asep-tically transferred to 1.5% non-nutrient agar (NNA) plates coated with a heat-inactivated *Escherichia coli* (ATCC 25922), at $10^8$ CFU/mL, 100 µL/plate to promote FLA growth [19]. After sealing with Parafilm, the plates were incubated at room temperature for up to 8 weeks. They were examined daily from week 2 using a light microscope at 100x magnification for screening and at 400x magnification for detailed examination.

After the initial microscopic identification, positive samples were sent to the Protozoology Laboratory at Shahid Beheshti University of Medical Sciences for a final microscopic identification and further analysis. The genus of the iso-lated FLAs was determined based on Page's taxonomic key and using the morphological characteristics of their tropho-zoites and cysts. [20]. Additionally, the positive plates were subcultured for axenic cultures. To achieve this, a fungal-free

agar block containing amoebae was cut and placed on a fresh medium to eliminate fungal contamination. This process was repeated until axenic cultures were obtained**.**

## Thermo- and osmo-tolerance tests

Thermo-tolerance and osmo-tolerance assays were conducted in accordance with established screening protocols to evaluate the potential pathogenicity of amoebae belonging to the genus *Acanthamoeba* [19,21]. For thermo-tolerance, cyst-saturated agar blocks were incubated at 37°C and 41°C; growth (outgrowth of trophozoites and/or increase in cell numbers) was monitored daily for 7 days by inverted microscopy at 400×. For osmo-tolerance, NNA plates supplemented with 0.5 M or 1.0 M D-mannitol were prepared; cyst-saturated agar blocks were placed centrally and incubated at room temperature. Growth was scored qualitatively as "-" (no growth), "1+" (rare/sparse growth), "2+" (moderate growth), "3+" (robust growth) based on the area and trophozoite density after 7 days, following criteria adapted from prior studies.

## Genomic DNA extraction and PCR amplification

The samples were prepared as required before extraction. To acquire appropriate samples, trophozoites and cysts were recovered from the surface of the NNA plate by scraping and then suspended in sterile phosphate-buffered saline (PBS) with a pH of 7.2. Then, centrifugation at 500 g for 5 minutes was performed, and the sedimented pellets were used for DNA extraction. Finally, the genomic DNA was extracted using the Favorgen commercial DNA extraction kit (Favorgen, Taiwan), following the manufacturer's recommended instructions. The DNA was stored at −20°C for future analysis.

PCR assays were conducted using genus-specific primers to target three genera of free-living amoebae (FLAs): *Acanthamoeba* (primers JDP1 and JDP2), Vahlkampfiids (primers ITS1 and ITS2), and *Vermamoeba* (primers Hv1227F and Hv1728R) (Table 1) [20,22,23]. The amplification protocol began with an initial denaturation step at 94°C for 5 minutes. This step was followed by 35 cycles, each consisting of denaturation at 94°C for 45 seconds, annealing at 56°C for 45 seconds, and extension at 72°C for 1 minute. The final extension took place at 72°C for seven minutes. To visualize the products, 5 µL of each reaction was separated on a 1.5% agarose gel stained with SafeStain (SinaClon, Iran) and screened under UV light.

## Sequencing and phylogenetic analysis

Fourteen PCR products of the 18S rRNA gene were successfully sequenced by the Noor Genetic Center in Tehran, Iran. Using Chromas software, the sequences were trimmed and edited against reference sequences of each genotype within *Acanthamoeba* sp. (Accession numbers: MH620477.1 and PP126062.1), *Vahlkampfia* sp. (Accession numbers: MF187528.1 and OL415191.1), and *V. vermiformis* (Accession numbers: DQ190245.1 and KY476315.1). A phylogenetic tree was obtained using Mega 5.1 software and the Maximum Likelihood algorithm to validate the taxonomic status of isolated FLAs. The topology of the constructed tree was supported by bootstrap values exceeding 70%. The accuracy of the phylogenetic tree was assessed through 1000 bootstrap resamplings, with a distance scale of 0.1 indicating the number

**Table 1. Primer sets for *Acanthamoeba*, *Vahlkampfiidae* and and *Vermamoeba*.**

| The genus | The name of Primer | The targeted gene | The sequences of primers (5′ to 3′) | The expected size (bp) of products | Reference |
|---|---|---|---|---|---|
| *Acanthamoeba* sp. | JDP1 and 2 | 18S rDNA | GGCCCAGATCGTTTACCGTGAA TCTCACAAGCTGCTAGGGAGTCA | ˷423–551 | [20] |
| Vahlkampfiids | ITS 1 and 2 | Ribosomal Internal Transcribed Spacer (ITS) | GAACCTGCGTAGGGATCATTT TTTCTTTTCCTCCCCTTATTA | ˷400–700 | [22] |
| *Vermamoeba* | Hv1227F and Hv1728R | 18S rDNA | TTACGAGGTCAGGACACTGT GACCATCCGGAGTTCTCG | ˷500 | [23] |

of base substitutions per site. *Naegleria* sp. (Accession number: FJ475126) and *V. vermiformis* were used as the outgroup branch.

## Results

### Cultivation and microscopic examination

A total of 10 samples (22.22%) out of 45 were found positive for FLAs using the microscopic method. One sample was contaminated with a single genus of FLAs, while the others exhibited mixed contamination with two or three genera (Table 2). The isolated organisms were *Vermamoeba* [9], *Acanthamoeba* [7], and *Vahlkampfia* [7] (Fig 1). Among the positive samples, 7 were obtained from the ophthalmology hospital and three from the general hospital. Additionally, 4 isolates were sourced from equipment, 5 from environmental samples, and 1 isolate was derived from surgical gowns (Table 2).

### Pathogenicity tests

The majority of isolated amoebas from the *Acanthamoeba* genus demonstrated osmo- and thermo-tolerance. Notably, two isolates demonstrated sensitivity to osmolarity, whereas the remaining isolates exhibited varying degrees of resistance to both osmolarity and heat shock. Specifically, Thermo-tolerance ranged from + at 37°C to 2+ at 41°C, while Osmo-tolerance ranged from – at 0.5 M to 1+ at 1 M (Table 3).

**Table 2. Source, microscopic, and molecular results of FLAs isolated from two hospitals in East Azerbaijan Province, Iran ((where "code" denotes the sample number).**

| | Code | Source | Microscopic Examination | | | Molecular examination | | |
|---|---|---|---|---|---|---|---|---|
| | | | *Acanthamoeba* | *Vahlkampfia* | *Vermamoeba* | *Acanthamoeba* | *Vahlkampfia* | *Vermamoeba* |
| Ophthalmology hospital | 1 | Ventilator | + | + | – | + | – | – |
| | 5 | Bed | + | – | + | + | – | – |
| | 7 | Bed | + | + | + | + | + | + |
| | 9 | Prob of laser | + | + | + | + | + | + |
| | 16 | Bed | – | + | + | – | + | + |
| | 23 | Forceps | + | – | + | + | – | + |
| | 24 | Scissors | – | + | + | – | – | – |
| General hospital | 31 | Urinary catheterization | + | + | + | – | + | + |
| | 35 | Angiocatheters | + | + | + | – | + | + |
| | 37 | Surgical gowns | – | + | – | – | – | – |

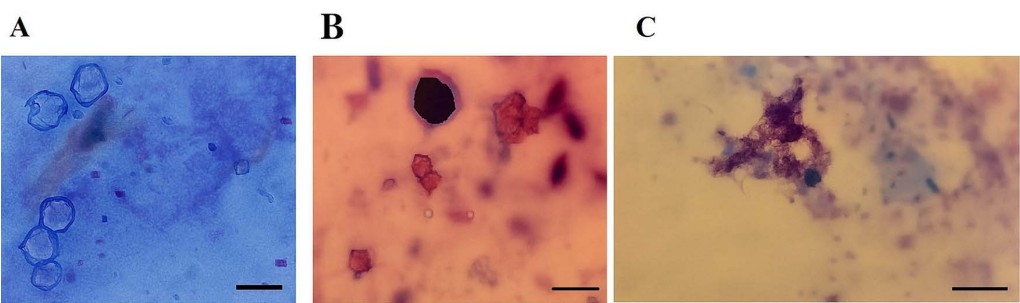

**A** **B** **C**

**Fig 1. Giemsa staining of A)** *Acanthamoeba* **sp., B) Vahlkampfiidae, C)** *V. vermiformis***. The scale bar is 10 µm.**

**Table 3. Results of Osmo- and Thermo-tolerance Tests for *Acanthamoeba* Isolates.**

| Code | Osmo-tolerance | | Thermo-tolerance | |
|------|----------------|---------------|------------------|----------------|
|      | Growth at 0.5 M | Growth at 1 M | Growth at 37°C | Growth at 41°C |
| 1  | 2+ | –  | 3+ | 1+ |
| 5  | +  | –  | 1+ | –  |
| 7  | 3+ | 1+ | 2+ | 2+ |
| 9  | –  | –  | 2+ | –  |
| 23 | 2+ | 1+ | 3+ | 1+ |
| 31 | 3+ | 1+ | 2+ | 2+ |
| 35 | –  | –  | 1+ | –  |

## Molecular detection and phylogenetic analysis

All microscopically positive samples were subjected to PCR, resulting in the amplification of desired bands in 16 samples, including 6 *Vermamoeba*, 5 *Acanthamoeba*, and 5 *Vahlkampfia*.

Based on sequence analyses of 18S rRNA, three genera of FLA, including *Acanthamoeba* sp. T4 genotype (n:5), *Vahlkampfia* sp. (n:3), and *V. vermiformis* (n:6) were identified, and 14 sequences were submitted to the GeneBank (NCBI) database (Table 4). The topology of the constructed phylogenetic tree indicated that each identified genus had been placed in its distinct clade. The submitted accession numbers of *Acanthamoeba* sp. T4 genotype, *Vahlkampfia* sp., and *V. vermiformis* are marked by an asterisk symbol (*) in Fig 2.

## Discussion

The results of the present study revealed the presence of free-living amoebae (FLAs), including potentially pathogenic genera, in the two hospitals studied. These microorganisms are common in hospital environments and should be considered significant health risks, particularly for individuals at high risk. Additionally, pathogenicity tests suggested that most *Acanthamoeba* isolates were resistant to osmolarity and heat shock, demonstrating potential virulence. This finding is

**Table 4. Free-Living Amoebae (FLA) Isolated from Hospital Equipment and Surfaces, Accompanied by GenBank Accession Numbers.**

| Code | Source | The identified FLA | The GenBank accession number |
|------|--------|--------------------|------------------------------|
| 1  | Ventilator | *Acanthamoeba* | PV545328.1 |
| 5  | Bed | *Acanthamoeba* | PV545391.1 |
| 7  | Bed | *Acanthamoeba* | PV545352.1 |
| 7  | Bed | *Vermamoeba vermiformis* | PV545494.1 |
| 9  | Prob of laser | *Acanthamoeba* | PV545393.1 |
| 9  | Prob of laser | *Vahlkampfia* | PV545397.1 |
| 9  | Prob of laser | *V. vermiformis* | PV545501.1 |
| 16 | Bed | *Vahlkampfia* | PV545396.1 |
| 16 | Bed | *V. vermiformis* | PV545561.1 |
| 23 | Forceps | *Acanthamoeba* | PV545395.1 |
| 23 | Forceps | *V. vermiformis* | PV545564.1 |
| 31 | Urinary catheterization | *Vahlkampfia* | PV562243.1 |
| 31 | Urinary catheterization | *V. vermiformis* | PV545562.1 |
| 35 | Angiocatheters | *V. vermiformis* | PV562244.1 |

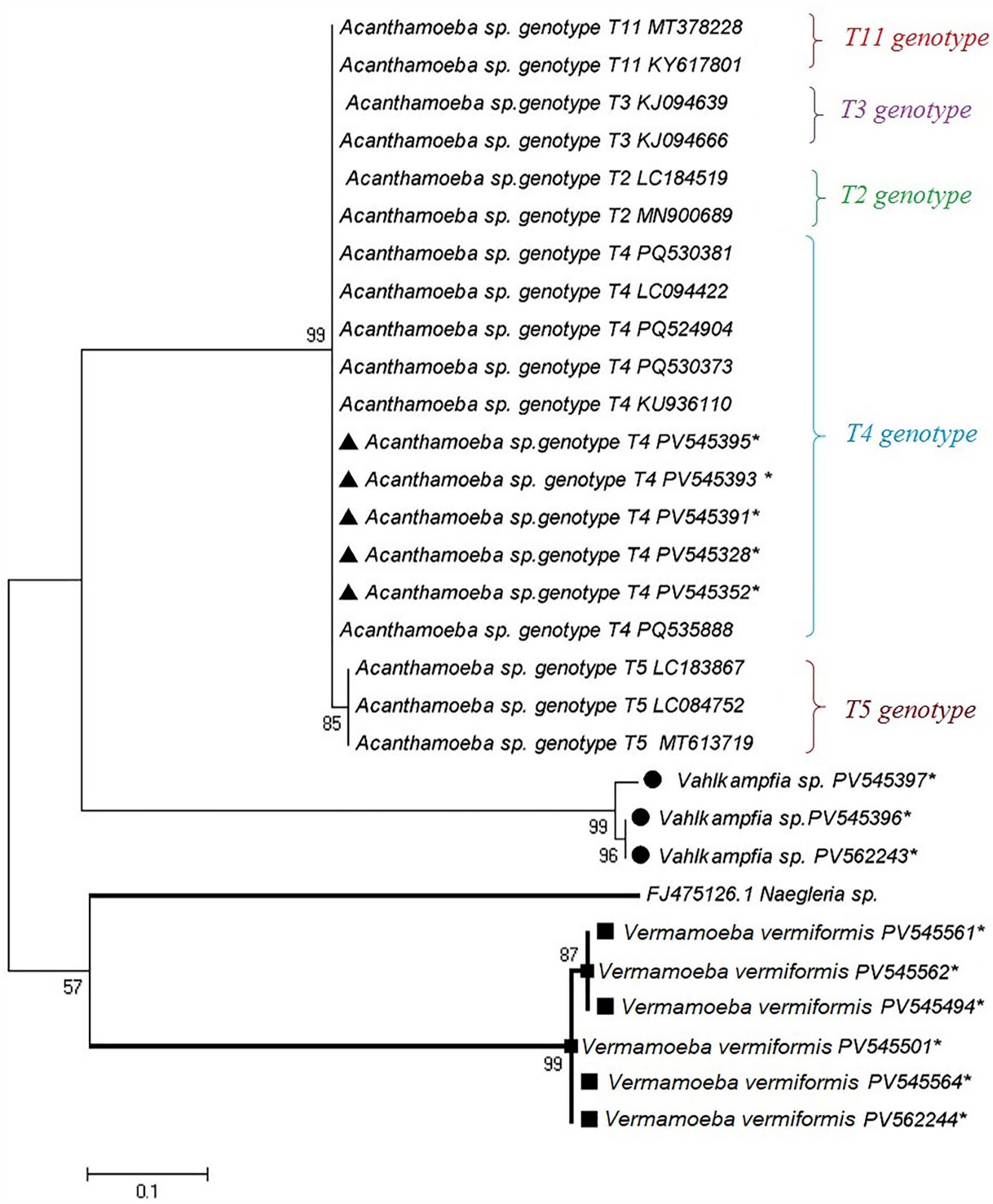

**Fig 2. A. Phylogenetic tree of *Acanthamoeba* sp. T4 genotype, *Vahlkampfia* sp. and *V. vermiformis* based on the 18S rRNA sequences.** The tree was generated by using the maximum likelihood algorithm and Kimura2-parameter mode. The sequences obtained from this study are shown by asterisk symbol (*)/geometric shapes. Based on sequence analyses of 18S rRNA, three genera of FLA including *Acanthamoeba* sp. T4 genotype (n:5), *Vahlkampfia* sp. (n:3) and *V. vermiformis* (n:6) were identified.

especially troubling, since *Acanthamoeba* causes serious infections like AK that can result in loss of vision, particularly among contact lens users and patients with previous corneal trauma. On the other hand, the pathogenicity tests indicated that most *Acanthamoeba* isolates were resistant to osmotic stress and heat shock, suggesting a potential virulence. This potential is particularly concerning as *Acanthamoeba* infections can lead to severe complications, including loss of vision among contact lens users and individuals with previous corneal trauma [18]. Therefore, implementing effective infection control measures, such as regular environmental monitoring and strict sterilization protocols, is critical to mitigate these risks.

The 22.22% prevalence of FLAs in our study is consistent with previous research findings. Niyyati et al. (2018) reported a 28% occurrence of FLAs in ICU nasal swabs and hospital dust in Iran, indicating a similar level of contamination despite the different types of samples examined [14]. A review study also indicates the presence of *Acanthamoeba* in the water systems, ventilation systems, and equipment of hospitals worldwide [13]. These studies, along with the results of the present study, suggest a consistent presence of FLAs in various hospital environments. Regarding the isolated genotypes of *Acanthamoeba*, the dominance of T4 in both environmental and clinical samples has been indicated in most previous studies, which is consistent with the results of the present study [24]. These consistencies suggest that FLAs, particularly *Acanthamoeba* T4, exhibit ecological adaptability across diverse hospital settings, potentially due to their resilient cyst forms that enable survival in suboptimal conditions.

The presence of FLAs in healthcare environments has been previously documented, although studies focusing on ready-to-use medical instruments are less common. For example, several studies reported the presence of FLAs, including *Acanthamoeba*, in dust from the hospital environment, hospital water systems, hospital air conditioning systems, or dialysis equipment previously [25–27]. However, our study is among the few to specifically target a broad range of medical instruments in active use, extending the known scope of FLA contamination beyond water systems and biofilms. The detection of *Vahlkampfia* and *Vermamoeba* alongside *Acanthamoeba* aligns with reports from environmental surveys, but their presence on surgical tools and patient-contact equipment appears novel and warrants further investigation. To our knowledge, this is the first study in Iran to document FLAs on a wide range of hospital equipment, providing valuable data on the regional epidemiology of these organisms.

The present investigation documented significant contamination with FLAs across both ophthalmology and general hospital environments. The detection of *Acanthamoeba*, *Vahlkampfia*, and *Vermamoeba* genera in critical medical equipment such as ventilators, laser probes, surgical instruments, and catheters represents a noteworthy finding with substantial clinical implications. The detection of potentially pathogenic FLAs in hospital environments and medical instruments carries significant clinical implications. *Acanthamoeba* species, which had significant prevalence in our samples, are known causative agents of AK and granulomatous amoebic encephalitis (GAE). The presence of these organisms in ophthalmological equipment is particularly concerning, given the potential for direct contact with the eyes. Our results indicate that FLAs, particularly *Acanthamoeba*, are resistant to routine cleansing due to their cyst form. Consequently, more effective disinfection methods should be employed for equipment that has direct contact with sensitive patient areas, particularly surgical instruments used in eye surgery. A recent study indicates that octenidine dihydrochloride (OCT) and quaternary ammonium compounds (QACs) are two commercially available agents effective in eliminating *Acanthamoeba* trophozoites and cysts when applied for an appropriate contact duration [21]. These substances are suitable for both skin and superficial applications in human, and their effectiveness is markedly affected by factors such as concentration (for example, cyst-specific EC50 values), duration of exposure, and the particular formulation used. In applied settings, chemical disinfectants may show diminished activity when contact time is insufficient, the active ingredient is below the optimal concentration, or organic material and biofilms are present, as these can shield microorganisms or neutralize biocidal agents. Although autoclaving at 121°C for 15 minutes is generally recommended as a standard sterilization procedure [2], experimental evidence indicates that this regimen may not invariably eliminate all *Acanthamoeba* cysts, highlighting the necessity of validating sterilization protocols for highly resistant protozoa. Accordingly, ongoing surveillance is required,

since amoebae can be reintroduced and re-establish contamination on instruments between autoclaving cycles and subsequent use, particularly in environments prone to biofilm development and organic soiling on device surfaces.

The isolation of FLAs from hospital environments and surgical equipment is critically important due to their symbiotic association with pathogenic bacteria, including *Legionella* and *Pseudomonas* [28]. The presence of these bacteria in healthcare settings poses significant risks, particularly for patients who are immunocompromised patients who are vulnerable to infections. Furthermore, contamination of surgical equipment can facilitate the direct transmission of free-living amoebae and their associated bacteria into a patient's body during invasive procedures, such as surgery.

Another notable finding in this study was the discrepancy between microscopic and molecular examination results. This result can be due to various reasons, including the genetic diversity of different isolates, the ineffectiveness of genus-specific primers, the presence of PCR inhibitors, the failure of DNA extraction due to the resistance of the cyst wall, or inaccuracies in microscopic diagnosis, which can lead to the use of incorrect primers [29–32]. Intrinsic variation in assay performance may also play a role, because PCR identifies target nucleic acid sequences regardless of organism viability and can therefore amplify DNA derived from non-viable cells that would not be detected by culture or other viability-dependent methods. In contrast, culture-based and similar techniques detect only viable organisms, so a low microbial burden, demanding growth requirements, or a viable-but-nonculturable state can lead to negative culture or microscopic results even when pathogen DNA is still detectable. Such discrepancies highlight the importance of employing complementary diagnostic approaches to ensure accurate identification, particularly in clinical contexts where precise species identification can inform risk assessment.

Based on the phylogenetic tree constructed, the isolated FLAs from the two hospitals clustered into three distinct genera: *Acanthamoeba* sp. (T4 genotype), *Vahlkampfia* sp., and *V. vermiformis*. The presence of *the Acanthamoeba* T4 genotype is consistent with previous reports identifying this genotype as the most prevalent and clinically significant genotype associated with human infections, including keratitis and other opportunistic infections [20]. The phylogenetic placement of the isolates alongside reference sequences from GenBank confirms their taxonomic affiliations and supports the reliability of the molecular identification approach used. The clustering pattern suggests that these FLAs share evolutionary relationships with environmental and clinical isolates reported worldwide, indicating their widespread distribution and ecological adaptability.

To build on these findings, future research should prioritize enhanced surveillance programs in Iranian hospitals to monitor FLA contamination over time, including longitudinal studies across multiple facilities. Validation using culture-independent methods, such as metagenomic sequencing, would provide a more comprehensive view of microbial communities and overcome limitations in traditional culturing and PCR approaches. Such efforts could inform targeted interventions and policy updates aimed at improving infection control.

## Conclusion

This study identified *Acanthamoeba*, *Vahlkampfia*, and *Vermamoeba* on ready-to-use medical instruments in two Iranian hospitals. The detection of potentially pathogenic *Acanthamoeba* highlights the need for enhanced infection control measures to protect vulnerable patients. These findings contribute to the existing body of knowledge regarding FLA contamination in healthcare environments, highlighting the importance of monitoring and improving sterilization processes for medical equipment to mitigate the risk of nosocomial infections.

### Limitations

The present study has some limitations that must be mentioned. The first limitation is the small sample size and limited sampling locations, which could affect the generalizability of the results; therefore, it is recommended that other similar studies be conducted. Furthermore, in this and similar studies, FLA detection relies solely on cultivation methods paired with specific molecular assessments applied to positive cultures, potentially leading to the loss of uncultivable or

low-abundance amoebae. The absence of culture-free techniques, such as metagenomic analysis, represents a significant drawback, as these strategies could provide a more comprehensive and precise depiction of microbial populations in healthcare facilities. Despite these limitations, this study provides valuable information on the prevalence of FLA in Medical Devices and Hospital Environments.

## Supporting information

**S1 Table. Exact source of samples.**
(DOCX)

## Acknowledgments

The authors sincerely thank the administrators, authorities, and staff of the Sarab Faculty of Medical Sciences and the two participating hospitals for their cooperation and assistance during sample collection. The authors also appreciate the use of language-support AI tools during manuscript preparation to improve clarity and readability. In addition, they acknowledge the statistical support provided by the Clinical Research Development Unit of Al-Zahra Hospital, Tabriz University of Medical Sciences.

## Author contributions

**Conceptualization:** Hamed Behniafar, Maryam Niyyati.

**Data curation:** Ali Bahadori, Leili Valizadeh, Fatemeh Mahdavi.

**Funding acquisition:** Hamed Behniafar.

**Investigation:** Hamed Behniafar, Ali Bahadori, Soghra Valizadeh, Hosein Pazoki, Fatemeh Mahdavi.

**Methodology:** Hamed Behniafar, Maryam Niyyati.

**Software:** Adel Spotin, Hosein Pazoki.

**Visualization:** Fatemeh Mahdavi.

**Writing – original draft:** Hamed Behniafar, Soghra Valizadeh, Leili Valizadeh.

**Writing – review & editing:** Hamed Behniafar, Maryam Niyyati.

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
