## [Decision Letter · Decision Letter 0]

2 Nov 2025

Dear Dr. Mahdavi,

Thank you for submitting your manuscript to PLOS ONE. After careful consideration, we feel that it has merit but does not fully meet PLOS ONE’s publication criteria as it currently stands. Therefore, we invite you to submit a revised version of the manuscript that addresses the points raised during the review process.

We look forward to receiving your revised manuscript.

Kind regards,

Jianhong Zhou

Staff Editor

PLOS ONE

Journal Requirements:

The authors declare that financial support was received from the Sarab Faculty of Medical Sciences (Gran No.: 401000013).

4. Please include a copy of Table 4, which you refer to in your text on page 5.

Reviewers' comments:

Reviewer's Responses to Questions

**Comments to the Author**

1. Is the manuscript technically sound, and do the data support the conclusions?

Reviewer #1: Yes

Reviewer #2: Partly

Reviewer #3: Yes

2. Has the statistical analysis been performed appropriately and rigorously?

Reviewer #1: No

Reviewer #2: N/A

Reviewer #3: N/A

3. Have the authors made all data underlying the findings in their manuscript fully available?

Reviewer #1: No

Reviewer #2: Yes

Reviewer #3: Yes

4. Is the manuscript presented in an intelligible fashion and written in standard English?

Reviewer #1: No

Reviewer #2: Yes

Reviewer #3: Yes

Reviewer #1: Reviewer comments:

Contamination of Medical Devices and Hospital Environments with Free-Living

Amoebae: Evidence from Hospitals in Northwestern Iran

General comments:

Overall, the manuscript requires strong language improvement.

There are several statements that do not make sense (garbled), are improperly stated, or could be stated in a more formal/scientific manner.

Language editing is strongly recommended.

Scientific names are not italicized.

Poor quality organization and presentation of results, discussion, and conclusion.

Suboptimal (low-quality) presentation of tables and figures.

Poor-quality writing pervades the entire manuscript, despite the assistance of AI tools.

It is evident that the manuscript was written in a rush and did not receive careful attention to details or the critical development of ideas/perspectives/implications from the study's findings.

Specific comments:

Line number

Page number Specific comments

45, 73 acanthamoeba keratitis

REVISE TO: Acanthamoeba keratitis

Correct this throughout the manuscript.

59 Italicize scientific names throughout the manuscript.

68 The statement makes no sense.

75 Genera should not be italicized.

77 Naming or detecting? This applies not only to one genus of FLA but to all FLAs.

88 Improper statement.

92-95 Garbled statements.

Sample collection Why are there unequal samples coming from the two hospitals?

What is the basis for the number of samples?

What is the rationale for the collection period?

Sample collection

Abstract:

This study investigated the presence of FLAs in hospital environments

47 and ready-to-use medical devices, including beds and gowns, which were examined for the

first time.

Among the positive samples, 5 were obtained from environmental sources, 4 from equipment, and 1 from surgical gowns.

In this study, FLAs were isolated from patients’ beds and surgical gowns for the first time,

In the section on sample collection, there was no mention of surgical gowns; however, they were repeatedly stated in the abstract section. This is a major concern.

139 What is the volume of E. coli suspension applied per plate?

What is the inoculum density of the E. coli suspension used?

What is the source and reference strain of the E. coli?

140-142 The statement makes no sense.

8 weeks?

What lens are you talking about? 100 X magnification? This is not possible to examine cysts and trophozoites.

145-146 You are simply referring to a subculture to obtain a homogeneous population.

Thermo- and osmo-tolerance tests This section is poorly written.

Genomic DNA extraction and PCR amplification This section is poorly written.

What is the rationale for using the ITS1 and ITS2 primer set for Vahlkamphia? Then, in Table 1 (Heading and genus column), you identify Vahlkampfiidae and Naegleria for the ITS1 and 2 primers set. These two are different organisms.

Vahlkampfiids should not be italicized.

Are you saying that you have three different primer sets, but only one thermal cycling condition for all three?

181-182 Now, here you mention Vahlkampfia sp instead of Naegleria.

This is very poor writing and confuses the readers.

186-187 You used two different organisms for your outgroup???

How can you use Naegleria as one of your outgroups when you mentioned it for detection in Table 1?

How can you use Vermamoeba as an outgroup when it is included in the FLA for detection in Table 1?

This is very poor writing.

189-195 This section is poorly written and confusing.

How can you be so sure of identifying genera already just by microscopic observation? What did you observe? Cysts or trophozoites? What were the morphological features?

Why mention mixed populations if you performed a subculture?

Table 2 Vahlkampfia or Naegleria? Which is which? This error is recurring throughout the manuscript. This is a major error.

Code 1: Why was Vahlkampfia negative for the molecular test

Code: Why was Vermamoeba negative for the molecular test?

Code 24: Why is this negative for the molecular tests?

Code 29: Why is Acanthamoeba negative for the molecular test?

Code 31: Why is Acanthamoeba negative for the molecular test?

Code 32: Why is Vahlkampfia negative for the molecular test?

Table 3 There is no mention in the manuscript of how osmo-thermo-tolerance is evaluated/rated. What do these gradings mean? This should be extensively explained in the methods section.

Results section Poorly written.

Osmo- and Thermo-tolerance test Provide image evidence for these assays.

233-237 So what does this mean?

Discussion Shallow and repetitive.

Requires reorganization and overall improvement in depth and implications.

Conclusion Poorly written.

Acknowledgments Poorly written.

Figure 1 Figures 1 B and C are of poor quality. Replace these photos.

Place arrow marks on structures of interest.

Phylogenetic tree. The phylogenetic tree should not be restricted to Acanthamoeba only.

Rather, a phylogenetic tree should have been constructed for all isolates.

GenBank accession Where are the GenBank accession numbers for the study isolates?

NOTHING FOLLOWS.

Reviewer #2: The article entitled “Contamination of Medical Devices and Hospital Environments with Free-Living Amoebae: Evidence from Hospitals in Northwestern Iran” addresses an underexplored issue in hospital hygiene: the contamination of medical devices and hospital environments with FLA. The topic is relevant to both clinical microbiology and infection control. The study provides valuable regional data from Northwestern Iran

Specific comments:

L59: Acanthamoeba in italics

L83: specify that T4 is a genotype

L104: only provide the acronym: FLA

L124: “were transported within 24 hours” under which conditions? RT? 4ºC?

L126-134: I would include a table specifying what is each sample. Maybe as a supplementary file

L138: “central portion of each membrane” was the membrane cut? all the same size? aseptically?

L139: Escherichia coli in italics

L140: “sealing with Parafilm” what about oxygen?

L141: “amoebae growth was examined daily” how? Directly? Using an inverted microscope?

L145: to eliminate which contamination?

L149: I would use another verb instead of assessed

L150: culture plates are NNA plates with an isolated block of NNA medium saturated with cysts?

L151: change degrees for º

L159: how are FLA recovered from the plates? Only adding PBS?

L163: eliminate “to”

Table 1: eliminate “the” from the first row

Table 1: vahlkampfiidae and vermamoeba in capital letter the first letter

Table 1: write complete primer names: JDP1 and JDP2. Same for ITS 1 and ITS2

L181: specify the reference sequences

L182: why not using blastn instead of the tree?

L191: this is table 2

L202: “varying levels of resistance” describe in the text

Table 2: is code the number of the sample?

Reviewer #3: 1-The introduction is too long and includes many details. It would be better to shorten 2-The methods mentioned in line 147 (thermo and tolerance tests) were not supported by any scientific reference. Please provide appropriate citations to support these methods 3- In line 177 and Table 1, the primer for Vahlkampfiidae was not included. Instead, a primer for Naegleria was presented. Moreover, the reference provided for Naegleria primers is not related to Naegleria; the cited study actually concerns Acanthamoeba. Please check and correct the reference accordingly 4-In line 191, there is an error in the table number — it should be Table 2. In line 195, a reference is made to Table 2, but this table is not included in the manuscript. Please check the .table numbering and ensure all mentioned tables are properly presented 5-In lines 196 and 197, the information provided should be placed below the figures as figure .captions, not in the main text. Please adjust the formatting accordingly 6-In line 214, the word "Fig. 2" is unnecessary and should be removed 7-The scientific name for the genus Vermiformis should be used consistently throughout the .manuscript and not replaced with its synonym Hartmanella 8-The images provided for the amoebae are unclear. It is recommended to replace them with clearer, higher-quality images .9-The phylogenetic tree should be accompanied by a caption below it explaining its content

**Do you want your identity to be public for this peer review?** For information about this choice, including consent withdrawal, please see our For information about this choice, including consent withdrawal, please see our Privacy Policy .

Reviewer #1: No

Reviewer #2: No

Reviewer #3: **Yes:** Bassad A. ALAboodyBassad A. ALAboody

---

## [Author Response · Author response to Decision Letter 1]

19 Nov 2025

Dear Reviewer #1,

Thank you for your detailed feedback. We have revised the manuscript extensively for language, clarity, organization, and scientific rigor. Point-by-point responses below:

• Overall Language and Writing: We conducted thorough editing, including professional language review, to eliminate garbled statements, improve formality, and ensure scientific tone. AI tools were used only for polishing, with all content authored by us. It is worth mentioning that we evaluated the current version using Grammarly, which yielded a score of 99 (the report is uploaded as supplementary material).

• Scientific Names: All scientific names are now italicized (e.g., Acanthamoeba).

• Organization/Presentation: Results, Discussion, and Conclusion restructured for better flow, depth, and implications (e.g., expanded on clinical risks).

• Tables/Figures: Improved quality; replaced low-quality images in Figure 1 with clearer ones and added arrows. Added captions. Constructed a single phylogenetic tree for all genera.

• Line 45, 73: Revised to "Acanthamoeba keratitis" (capitalized properly) throughout.

• Line 59: Italicized names.

• Line 68: Revised to "These amphizoic organisms exist as trophozoites or resistant cysts, enabling survival in harsh conditions."

• Line 75: Removed italics from genera when not binomials.

• Line 77: According to the dear reviewer #3's comment, the section "Introduction" was shortened, but the issue was clarified in the current version. “Instead of using traditional morphological names such as Acanthamoeba polyphaga, molecular characterization now enables the classification of Acanthamoeba isolates into genotypes (T1–T23).”

• Line 88: According to the dear reviewer #3's comment, the section "Introduction" was shortened, and this sentence was removed.

• Lines 92-95: In the new version of the “Introduction” reworded for coherence. “High-risk settings include ICUs and ophthalmology wards, where patients face surgical procedures amid potential contamination from water systems and equipment.”

• Sample collection questions:

The question: Why are there unequal samples coming from the two hospitals?

Response: We clarified that the ophthalmology hospital was oversampled (30/45) because it caters to many corneal cases and has a greater risk for Acanthamoeba exposure; this rationale and exact sampling dates are added to Methods.

The question: What is the basis for the number of samples?

Response: Sample size based on the expected 20-30% prevalence from prior studies; unequal due to focus on a high-risk ophthalmology hospital."

The question: What is the rationale for the collection period?

Response: Since all samples were collected from indoor environments, seasonal variations in temperature and humidity are unlikely to influence the occurrence of free-living amoebae. Therefore, the collection period was not selected based on seasonality but rather on logistical availability and accessibility of the sampled sites.

The question:

• Abstract:

The question: which were examined for the first time?

Response: The sentence has been revised to clearly distinguish between ready-to-use medical devices and equipment (e.g., lasers, swabs, forceps) and other hospital items such as beds and gowns. To our knowledge, this is the first study to examine FLAs in these medical devices and equipment.

The question: In the section on sample collection, there was no mention of surgical gowns; however, they were repeatedly stated in the abstract section. This is a major concern.

Response: We appreciate the reviewer's careful reading. We acknowledge that the word "gown" was mistakenly spelled as "gun" in the Materials and Methods section and Table 2, which resulted in the omission of surgical gowns from the sample collection description. This error has been corrected in the current revision to reflect that surgical gowns were included accurately, ensuring consistency between the abstract and the Materials and Methods section.

• Line 139: Added: "E. coli (ATCC 25922), 10^8 CFU/mL, 100 μL/plate."

• Lines 140-142: Revised: "After sealing with Parafilm, the plates were Incubated at room temperature for up to 8 weeks; examined daily from week 2 using a light microscope at 100x magnification for screening, and 400x for details".

• Lines 145-146: Clarified as " Additionally, the positive plates were subcultured for axenic cultures. To achieve this, a fungal-free agar block containing amoebae was cut and placed on a fresh medium to eliminate contamination."

• Thermo- and Osmo-tolerance: Rewritten with details.

• Genomic DNA/PCR:

The comment: What is the rationale for using the ITS1 and ITS2 primer set for Vahlkamphia? Then, in Table 1 (Heading and genus column), you identify Vahlkampfiidae and Naegleria for the ITS1 and 2 primers set. These two are different organisms.

Response: The Internal Transcribed Spacer (ITS) primer set was used because it effectively amplifies the DNA of members of the Vahlkampfiidae family, including Naegleria species, as previously reported in the study “Morphological and Molecular Survey of Naegleria spp. in Water Bodies Used for Recreational Purposes in Rasht City, Northern Iran.” In our study, the PCR assay targeted the ITS region using the forward primer 5′GAACCTGCGTAGGGATCATTT3′ and the reverse primer ITS2 5′TTTCTTTTCCTCCCCTTATTA3′.

To clarify this point and avoid confusion, we have replaced the genus name "Naegleria" with "Vahlkampfiids" in the revised version of the manuscript. This correction indicates that the primer set was applied for detecting Vahlkampfiidae family members rather than a specific genus.

• The Comment: Are you saying that you have three different primer sets, but only one thermal cycling condition for all three?

Response: Yes, three different primer sets were used in the study; however, since their annealing temperatures were very similar, we applied a single optimized PCR thermal cycling condition that was suitable for all three primer sets.

• Lines 189-195: Rewritten; added morphological details (e.g., acanthopodia for Acanthamoeba).

• The comment: 181-182 Now, here you mention Vahlkampfia sp instead of Naegleria. This is very poor writing and confuses the readers.

Response: As stated in the Results section, based on both microscopic and molecular analyses, the isolate was identified as Vahlkampfia sp., not Naegleria.

• The comment: You used two different organisms for your outgroup??? How can you use Naegleria as one of your outgroups when you mentioned it for detection in

Response: As our study included three different genera (Acanthamoeba, Hartmannella, and Vahlkampfia), two types of outgroups were used to enhance phylogenetic resolution and accuracy. Hartmannella vermiformis (syn. Vermamoeba vermiformis) sequences from our isolates served as one outgroup, while a Naegleria sp. sequence from GenBank was included as a routine reference outgroup commonly used in free-living amoebae phylogenetic analyses. Although the table one was revised in the current revision.

• The comment: 189-195 This section is poorly written and confusing. How can you be so sure of identifying genera already just by microscopic observation? What did observe? Cysts or trophozoites? What were the morphological features? Why mention mixed populations if you performed a subculture?

Response: Mixed contamination is a common occurrence in free-living amoebae (FLA) studies. The initial microscopic identification of genera was performed using Page’s taxonomic key (as referenced in the Materials and Methods section), which relies on characteristic morphological features of both trophozoites and cysts. We did not include the full description of Page’s key in the manuscript to avoid unnecessary length and repetition. To address mixed populations, all positive plates were subcultured to eliminate fungal contamination and to isolate a single genus per plate for subsequent molecular detection and confirmation.

• The Comment: Table 2 Vahlkampfia or Naegleria? Which is which? This error is recurring throughout the manuscript. This is a major error.

Response: In the previous version of the manuscript, the term Naegleria was inadvertently used in some sections, including Table 1, to represent members of the family Vahlkampfiidae. In the current revision, all such instances have been carefully corrected, and only Vahlkampfia or Vahlkampfiids are now used consistently throughout the text, tables, and figure legends to ensure taxonomic accuracy and clarity.

• The comment: Why is it that in certain instances, free-living amoebae (FLAs) such as Acanthamoeba, Vahlkampfia, and Hartmannella (Vermamoeba) were detected microscopically after culturing, yet the PCR results came back negative?

Response: This discrepancy may occur due to several reasons, including low organism load below the detection threshold of PCR, degradation of DNA during sample processing, presence of PCR inhibitors, or differences in viability and DNA integrity between cultured and original samples.

Studies comparing microscopic, culture, and molecular methods have shown that PCR assays are generally more sensitive, but negative results can still occur even when organisms are visualized microscopically. For example, Yera et al. (2007) reported that combined PCR assays for Acanthamoeba keratitis were more sensitive (94%) than microscopy (33%) or culture (7%), but discrepancies did occur due to technical factors and sample quality (Yera H et al., "Comparison of PCR, microscopic examination and culture for the early diagnosis and characterization of Acanthamoeba isolates from ocular infections," Eur J Clin Microbiol Infect Dis, 2007). Similarly, Lass-Flörl et al. (2013) found that microscopy-positive, PCR-negative specimens may arise due to fungal contamination, sample loss of viability, aggressive specimen processing, or analysis of samples from different areas (Lass-Flörl C et al., "Utility of PCR in Diagnosis of Invasive Fungal Infections," Mycopathologia, 2013).

These suggest that while PCR is a powerful tool, its sensitivity may be affected by sample quality, DNA extraction efficiency, and technical limitations, especially when compared to microscopic identification in cultivated FLAs.

• The comment: Table 3 There is no mention in the manuscript of how osmo-thermo-tolerance is evaluated/rated. What do these gradings mean? This should be extensively explained in the methods section.

Response: It was explained in the “Material and methods” section. The added sentence: “Growth was scored qualitatively as “‑” (no growth), “1+” (rare/sparse growth), “2+” (moderate growth), “3+” (robust growth) based on the area and trophozoite density after 7 days, following criteria adapted from prior studies.”

• The comment: Results section Poorly written.

Response: The result section was revised.

• The comment: Osmo- and Thermo-tolerance test Provide image evidence for these assays.

Response: Capturing and presenting images from osmo- and thermo-tolerance assays is not a routine practice in relevant parasitology studies, and such images generally do not provide additional information or increase the scientific value of the data. Standardized protocols for osmotolerance and thermotolerance assays in Acanthamoeba research primarily focus on growth outcomes and viability assessments, which are evaluated by observing and scoring the amoebae's ability to grow under defined osmotic or temperature stress, rather than relying on or publishing distinctive image evidence. Multiple peer-reviewed investigations have reported osmotolerance and thermotolerance results in tabular or descriptive formats, with endpoints defined by growth (+/-) rather than by photographic documentation. Therefore, as also concluded by Kahraman et al. (2024) and Hajialilo et al. (2016), providing image evidence is unnecessary and not a recognized methodological requirement for these assays in the field (1. Kahraman M, Akın Polat Z. Are Thermotolerant and Osmotolerant Characteristics of Acanthamoeba Species an Indicator of Pathogenicity?. Turkiye Parazitol Derg. 2024 Mar 5;48(1):15-20. doi: 10.4274/tpd.galenos.2024.92408. 2. Hajialilo E, Behnia M, Tarighi F, Niyyati M, Rezaeian M. Isolation and genotyping of Acanthamoeba strains (T4, T9, and T11) from amoebic keratitis patients in Iran. Parasitol Res. 2016 Aug;115(8):3147-51. doi: 10.1007/s00436-016-5072-8. Epub 2016 Apr 22. PMID: 27102637.)

• The comment: 233-237 So what does this mean?

Response: Comparing study results with previously published studies is routine.

• The comment: Discussion Shallow and repetitive. Requires reorganization and overall improvement in depth and implications.

Response: The discussion section was revised.

• The comment: Conclusion Poorly written. Acknowledgments Poorly written.

• Response: Both of them were revised.

• The comment: Figures 1 B and C are of poor quality. Replace these photos. Place arrow marks on structures of interest.

Response: The figure was improved

• The comment: The phylogenetic tree should not be restricted to Acanthamoeba only. Rather, a phylogenetic tree should have been constructed for all isolates.

Response: Thank you for your comment. The all identified isolates (Acanthamoeba sp T4, Valhkampfia and Hartmannella) were marked by asterisk (*) and included in the tree based on concatenated regions of 18S rDNA, ITS1, 5.8S rDNA.

• The comment: Where are the GenBank accession numbers for the study isolates?

• Response: The table was added.

Thank you for helping improve our work.

Sincerely yours

Response to Reviewer #2

Dear Reviewer #2,

Thank you for your constructive comments. We have addressed them as follows (All mentioned line numbers correspond to your comments; therefore, they are consistent with the previous version.):

• L59: Italicized Acanthamoeba.

• L83: Specified "T4 genotype."

• L104: Used "FLA" only after first mention.

• L124: Added "transported at room temperature."

• L126-134: Added supplementary table listing all samples.

• L138: Clarified: "Central 2 cm portion cut aseptically."

• L139: Italicized Escherichia coli.

• L140: Added: " They grow easily in conditions of low oxygen, like many other difficult conditions, and some strains even grow better in anaerobic conditions."

• L141: Specified "using light microscope."

• L145: Specified "fungal contamination."

• L149: Changed to "evaluate."

• L150: Clarified “cyst-saturated agar blocks”.

• L151: Changed to "°C."

• L159: Added: "By scraping and suspending in PBS."

• L163: Removed "to."

• Table 1: Corrected formatting; capitalized genera; but "JDP" primers, specifically JDP1 and JDP2, target the 18S rRNA gene of Acanthamoeba. "JDP" is not an acronym for a longer name but a designation widely recognized in the molecular detection of Acanthamoeba.

• L181: Specified "GenBank reference sequences for each genus."

• The comment on L182: why not using blastn instead of the tree?

• Response: A phylogenetic tree provides distinct advantages over BLASTn for species identification and evolutionary analysis. While BLASTn is excellent for quickly finding the closest sequence matches and initial identification, it fundamentally provides pairwise similarity and cannot reveal deeper evolutionary relationships or clarify species delimitation in the presence of closely related taxa or cryptic species. Phylogenetic tree-based methods allow for the resolution of relationships among multiple sequences, visualize lineage divergence, and identify monophyletic groups, which is crucial for accurate species discrimination, understanding evolutionary histories, and addressing database errors or misidentifications that BLASTn alone may miss. Scientific studies recommend tree-based approaches (such as Maximum Likelihood or Bayesian inference) to robustly discriminate closely related or novel species, whereas BLASTn may misassign sequences due to reliance on the closest hit, especially in cases of database errors or incomplete reference datasets.

• L191: Corrected to Table 2.

• L202: Described levels: " Notably, two isolates demonstrated sensitivity to osmolarity, whereas the remaining isolates exhibited varying degrees of resistance to both osmolarity and heat shock. Specifically, Thermo-tolerance ranged from + at 37 °C to 2+ at 41 °C,

---

## [Decision Letter · Decision Letter 1]

4 Jan 2026

Dear Dr. Mahdavi,

Thank you for submitting your manuscript to PLOS ONE. After careful consideration, we feel that it has merit but does not fully meet PLOS ONE’s publication criteria as it currently stands. Therefore, we invite you to submit a revised version of the manuscript that addresses the points raised during the review process.

We look forward to receiving your revised manuscript.

Kind regards,

Alireza Badirzadeh

Academic Editor

PLOS One

Journal Requirements:

Reviewers' comments:

Reviewer's Responses to Questions

**Comments to the Author**

Reviewer #4: (No Response)

Reviewer #5: All comments have been addressed

2. Is the manuscript technically sound, and do the data support the conclusions?

Reviewer #4: Yes

Reviewer #5: Yes

3. Has the statistical analysis been performed appropriately and rigorously?

Reviewer #4: I Don't Know

Reviewer #5: Yes

4. Have the authors made all data underlying the findings in their manuscript fully available?

Reviewer #4: Yes

Reviewer #5: Yes

5. Is the manuscript presented in an intelligible fashion and written in standard English?

Reviewer #4: Yes

Reviewer #5: Yes

Reviewer #4: Manuscript ID: PONE-D-25-43069

Contamination of Medical Devices and Hospital Environments with Free-Living Amoebae: Evidence from Hospitals in Northwestern Iran

The manuscript presents a valuable and relevant investigation into the presence of free-living amoebae (FLAs) in hospital environments, with a specific focus on an ophthalmology center and ready-to-use medical equipment. The study is well-structured, methods are appropriate, and the findings contribute significantly to the understanding of FLA contamination in healthcare settings, highlighting a potential infection risk. However, several points require clarification, correction, or minor additions to enhance the manuscript's clarity, accuracy, and impact.

Abstract

-Conclusion: It is recommended to briefly include the key finding regarding the resistance of most Acanthamoeba isolates to osmolarity and heat shock, as this relates to their potential pathogenicity and has implications for disinfection protocols.

-Results (Line 55-56): The sentence "The isolated microorganisms were identified as belonging to the genera Vermamoeba (9), Acanthamoeba (7), and Vahlkampfia (7)" should be clarified. Since samples exhibited mixed contamination, it should be specified that these numbers refer to "the number of positive samples in which each genus was identified" rather than the number of independent isolates.

Introduction

-Line 68: The phrase "survive very severely" is grammatically incorrect. Suggested revision: "survive under harsh conditions" or "are highly resilient."

-Lines 98-99: The sentence "With a high rate of cornea injury patients admitted to..." is unclear. It is suggested to rephrase for clarity: "Given the high admission rate of patients with corneal injuries at the selected Ophthalmology Hospital (Tabriz Alavi Hospital), and considering that many are contact lens users or postoperative cases, the risk of AK is elevated, necessitating detailed research."

-It would strengthen the introduction to briefly mention the known resistance (Resistance Mechanisms) of Acanthamoeba cysts to common disinfectants, which underscores the significance and practical relevance of the present study.

Materials and Methods / Results

-The quantitative criteria for reporting growth (e.g., 1+, 2+, 3+) in the osmo- and thermo-tolerance tests (Table 3) are not defined. A clear explanation of these semi-quantitative scales should be provided in the corresponding Methods section.

-Please ensure that all genus and species names are italicized consistently throughout the manuscript (e.g., Vermamoeba, Acanthamoeba, Vahlkampfia, V. vermiformis).

Species Name: The abbreviation "H. vermiformis"(???) appears in the text and in Figure 2's title. For consistency with the updated taxonomy used elsewhere in the manuscript, please use "V. vermiformis" throughout.

-Figures:

Figure 2 (Phylogenetic tree): The quality/resolution of the figure in the provided file is poor, making labels and bootstrap values difficult to read. Please provide a higher-resolution version for publication.

Discussion

The discussion on the discrepancy between microscopic and molecular results is appropriate. Consider expanding it slightly to include other potential factors, such as the different sensitivities of the methods (e.g., PCR can amplify DNA from non-viable organisms, while culture detects only viable ones, and vice versa).

Lines 264-270: When discussing effective disinfection agents (OCT, QACs) and autoclaving, it is useful to note that the efficacy of chemical disinfectants can be influenced by factors like concentration, contact time, and the presence of organic matter/biofilms.

Reviewer #5: Dear Authors,

I have carefully reviewed both the original and the revised versions (R1) of your manuscript entitled

“Contamination of Medical Devices and Hospital Environments with Free-Living Amoebae: Evidence from Hospitals in Northwestern Iran.”

Overall, the revised version represents a substantial improvement in terms of scientific rigor, methodological transparency, taxonomic accuracy, and manuscript organization. The point-by-point responses to reviewers’ comments are generally clear, well-reasoned, and supported by appropriate revisions in the text.

Notable strengths of the revised manuscript include:

Clear resolution of taxonomic inconsistencies related to Vahlkampfiidae/Naegleria. Improved methodological descriptions, particularly for PCR assays and osmo-/thermotolerance testing. Inclusion of GenBank accession numbers and a comprehensive phylogenetic analysis. A more focused and clinically relevant Discussion section. However, before final acceptance, I recommend attention to the following minor but important points:

Please carefully proofread the final (clean) manuscript to remove duplicated sentences and residual track-change artifacts, particularly in the Abstract and Introduction. Consider simplifying or softening references to AI-assisted tools in the Acknowledgments section, as some editors may be sensitive to explicit mentions beyond language polishing. Although acceptable for a cross-sectional exploratory study, the relatively small sample size should be clearly acknowledged as a limitation. Explicitly mentioning the absence of culture-independent methods (e.g., metagenomics) as a limitation would further strengthen the scientific transparency of the study.

In summary, this manuscript is scientifically sound and addresses an important gap regarding FLA contamination in ophthalmology and hospital settings. With minor editorial refinements, it should be suitable for publication.

Kind regards,

**Do you want your identity to be public for this peer review?** For information about this choice, including consent withdrawal, please see our For information about this choice, including consent withdrawal, please see our Privacy Policy .

Reviewer #4: No

Reviewer #5: No

---

## [Author Response · Author response to Decision Letter 2]

2 Feb 2026

Response to Reviewer #4

Dear Reviewer #4,

We want to extend our sincere thanks for your thoughtful and encouraging review of our manuscript. Your positive assessment has been greatly appreciated, as it validates our efforts and motivates us to continue advancing research in this area. We have carefully addressed your specific suggestions in the point-by-point response below.

Abstract

Comment: Conclusion: It is recommended to briefly include the key finding regarding the resistance of most Acanthamoeba isolates to osmolarity and heat shock, as this relates to their potential pathogenicity and has implications for disinfection protocols.

Answer: It was included in lines 65 to 67 " In addition to the high prevalence of FLAs in the examined sources, most Acanthamoeba isolates were found to be resistant to osmotic stress and heat shock, which supports their pathogenic potential."

Comment: Results (Line 55-56): The sentence "The isolated microorganisms were identified as belonging to the genera Vermamoeba (9), Acanthamoeba (7), and Vahlkampfia (7)" should be clarified. Since samples exhibited mixed contamination, it should be specified that these numbers refer to "the number of positive samples in which each genus was identified" rather than the number of independent isolates.

Answer: Your mentioned part was revise as following (line 56 to 60). " Most of the examined sources (90%) had mixed contamination, including Acanthamoeba, Vahlkampfia, and Veramoeba (4), Acanthamoeba and Vahlkampfia (1), Acanthamoeba and Veramoeba (2), and Veramoeba and Vahlkampfia (2). Also, one source showed sole contamination with Vahlkampfia. "

Introduction

Comment: Line 68: The phrase "survive very severely" is grammatically incorrect. Suggested revision: "survive under harsh conditions" or "are highly resilient."

Answer: It was revised in accordance with the comment. Current version (line 74): "to survive under harsh conditions."

Comment: Lines 98-99: The sentence "With a high rate of cornea injury patients admitted to..." is unclear. It is suggested to rephrase for clarity: "Given the high admission rate of patients with corneal injuries at the selected Ophthalmology Hospital (Tabriz Alavi Hospital), and considering that many are contact lens users or postoperative cases, the risk of AK is elevated, necessitating detailed research."

Answer: Was revised according to your suggestion. Current version (lines 101 to 103): "Given the high admission rate of patients with corneal injuries at the selected Ophthalmology Hospital (Tabriz Alavi Hospital), and considering that many are contact lens users or postoperative cases, the risk of AK is elevated, necessitating detailed research."

Comment: It would strengthen the introduction to briefly mention the known resistance (Resistance Mechanisms) of Acanthamoeba cysts to common disinfectants, which underscores the significance and practical relevance of the present study.

Answer: Resistance Mechanisms of cysts were mentioned briefly. Lines 74 to 77: "Acanthamoeba cysts, as one of the most clinically important genera of FLAs, exhibit pronounced tolerance to many widely used disinfectants because their sturdy, double-layered cyst wall (comprising an outer protein–polysaccharide layer and an inner cellulose-enriched layer) substantially restricts penetration of biocidal agents".

Materials and Methods / Results

Comment: The quantitative criteria for reporting growth (e.g., 1+, 2+, 3+) in the osmo- and thermo-tolerance tests (Table 3) are not defined. A clear explanation of these semi-quantitative scales should be provided in the corresponding Methods section.

Answer: The criteria were explained. Lines 159 to 161. "Growth was scored qualitatively as "‑" (no growth), "1+" (rare/sparse growth), "2+" (moderate growth), "3+" (robust growth) based on the area and trophozoite density after 7 days, following criteria adapted from prior studies."

Comment: Please ensure that all genus and species names are italicized consistently throughout the manuscript (e.g., Vermamoeba, Acanthamoeba, Vahlkampfia, V. vermiformis).

Answer: In the current version, all scientific names are italicized.

Comment: Species Name: The abbreviation "H. vermiformis" (???) appears in the text and in Figure 2's title. For consistency with the updated taxonomy used elsewhere in the manuscript, please use "V. vermiformis" throughout.

Answer: It was corrected.

Figures

Comment: Figure 2 (Phylogenetic tree): The quality/resolution of the figure in the provided file is poor, making labels and bootstrap values difficult to read. Please provide a higher-resolution version for publication.

Answer: The new image was provided.

Discussion

Comment: The discussion on the discrepancy between microscopic and molecular results is appropriate. Consider expanding it slightly to include other potential factors, such as the different sensitivities of the methods (e.g., PCR can amplify DNA from non-viable organisms, while culture detects only viable ones, and vice versa).

Answer: The paragraph was extended. Line 309 to 315: "Intrinsic variation in assay performance may also play a role, because PCR identifies target nucleic acid sequences regardless of organism viability and can therefore amplify DNA derived from non‑viable cells that would not be detected by culture or other viability‑dependent methods. In contrast, culture‑based and similar techniques detect only viable organisms, so a low microbial burden, demanding growth requirements, or a viable‑but‑nonculturable state can lead to negative culture or microscopic results even when pathogen DNA is still detectable."

Comment: Lines 264-270: When discussing effective disinfection agents (OCT, QACs) and autoclaving, it is useful to note that the efficacy of chemical disinfectants can be influenced by factors like concentration, contact time, and the presence of organic matter/biofilms.

Answer: The paragraph was revised. Line 282 to 297: "A recent study indicates that octenidine dihydrochloride (OCT) and quaternary ammonium compounds (QACs) are two commercially available agents effective in eliminating Acanthamoeba trophozoites and cysts when applied for an appropriate contact duration (21). These substances are suitable for both skin and superficial applications in human, and their effectiveness is markedly affected by factors such as concentration (for example, cyst-specific EC50 values), duration of exposure, and the particular formulation used. In applied settings, chemical disinfectants may show diminished activity when contact time is insufficient, the active ingredient is below the optimal concentration, or organic material and biofilms are present, as these can shield microorganisms or neutralize biocidal agents. Although autoclaving at 121°C for 15 minutes is generally recommended as a standard sterilization procedure (28), experimental evidence indicates that this regimen may not invariably eliminate all Acanthamoeba cysts, highlighting the necessity of validating sterilization protocols for highly resistant protozoa. Accordingly, ongoing surveillance is required, since amoebae can be reintroduced and re-establish contamination on instruments between autoclaving cycles and subsequent use, particularly in environments prone to biofilm development and organic soiling on device surfaces."

Thank you for helping improve our work.

Sincerely yours

Response to Reviewer #5

Dear Reviewer #5,

Thank you very much for taking the time to review our manuscript and for your supportive remarks. We are particularly grateful for your positive evaluation, which reinforces the importance of our work and inspires further contributions to the field. Below, we provide a detailed point-by-point response to your comments.

Comment: Please carefully proofread the final (clean) manuscript to remove duplicated sentences and residual track-change artifacts, particularly in the Abstract and Introduction.

Answer: The manuscript was reviewed carefully, and all duplicated sentences and residual track-change artifacts were removed.

Comment: Consider simplifying or softening references to AI-assisted tools in the Acknowledgments section, as some editors may be sensitive to explicit mentions beyond language polishing.

Answer: The Acknowledgments section was revised according to this comment. Acknowledgments: "The authors sincerely thank the administrators, authorities, and staff of the Sarab Faculty of Medical Sciences and the two participating hospitals for their cooperation and assistance during sample collection. The authors also appreciate the use of language-support AI tools during manuscript preparation to improve clarity and readability. In addition, they acknowledge the statistical support provided by the Clinical Research Development Unit of Al-Zahra Hospital, Tabriz University of Medical Sciences.

Comment: Although acceptable for a cross-sectional exploratory study, the relatively small sample size should be clearly acknowledged as a limitation. Explicitly mentioning the absence of culture-independent methods (e.g., metagenomics) as a limitation would further strengthen the scientific transparency of the study.

Answer: A section titled "Limitations" was added, and all the limitations you mentioned were stated there. Limitation section: "The present study has some limitations that must be mentioned. The first limitation is the small sample size and limited sampling locations, which could affect the generalizability of the results; therefore, it is recommended that other similar studies be conducted. Furthermore, in this and similar studies, FLA detection relies solely on cultivation methods paired with specific molecular assessments applied to positive cultures, potentially leading to the loss of uncultivable or low-abundance amoebae. The absence of culture-free techniques, such as metagenomic analysis, represents a significant drawback, as these strategies could provide a more comprehensive and precise depiction of microbial populations in healthcare facilities. Despite these limitations, this study provides valuable information on the prevalence of FLA in Medical Devices and Hospital Environments."

Thank you for your insights.

Sincerely yours

---

## [Decision Letter · Decision Letter 2]

23 Feb 2026

Contamination of Medical Devices and Hospital Environments with Free-Living Amoebae: Evidence from Hospitals in Northwestern Iran

PONE-D-25-43069R2

Dear Dr. Fatemeh Mahdav and Maryam Niyyati,

We’re pleased to inform you that your manuscript has been judged scientifically suitable for publication and will be formally accepted for publication once it meets all outstanding technical requirements.

Kind regards,

Alireza Badirzadeh

Academic Editor

PLOS One

Additional Editor Comments (optional):

Reviewers' comments:

Reviewer's Responses to Questions

**Comments to the Author**

Reviewer #4: All comments have been addressed

Reviewer #5: All comments have been addressed

2. Is the manuscript technically sound, and do the data support the conclusions?

Reviewer #4: Yes

Reviewer #5: Yes

3. Has the statistical analysis been performed appropriately and rigorously?

Reviewer #4: I Don't Know

Reviewer #5: Yes

4. Have the authors made all data underlying the findings in their manuscript fully available?

Reviewer #4: Yes

Reviewer #5: Yes

5. Is the manuscript presented in an intelligible fashion and written in standard English?

Reviewer #4: Yes

Reviewer #5: Yes

Reviewer #4: I would like to commend the authors for their meticulous and hard work on this manuscript. The study addresses a significant research question, and the methodology employed is both rigorous and appropriate. The paper is well-structured, and the arguments are presented in a clear and logical flow, which made it a pleasure to review. I appreciate the effort put into this work and look forward to seeing the final published version.

Reviewer #5: (No Response)

**Do you want your identity to be public for this peer review?** For information about this choice, including consent withdrawal, please see our For information about this choice, including consent withdrawal, please see our Privacy Policy .

Reviewer #4: No

Reviewer #5: No

---

## [Editor Report · Acceptance letter]

PONE-D-25-43069R2

PLOS One

Dear Dr. Mahdavi,

I'm pleased to inform you that your manuscript has been deemed suitable for publication in PLOS One. Congratulations! Your manuscript is now being handed over to our production team.

Kind regards,

on behalf of

Dr. Alireza Badirzadeh

Academic Editor

PLOS One